# Reaction of the Organisms of Young Football Players to City Smog in the Sports Training

**DOI:** 10.3390/ijerph17155510

**Published:** 2020-07-30

**Authors:** Henryk Duda, Łukasz Rydzik, Wojciech Czarny, Wiesław Błach, Karol Görner, Tadeusz Ambroży

**Affiliations:** 1Faculty of Physical Education and Sport, Institute of Sport, University of Physical Education in Krakow, 31-541 Kraków, Poland; henryk.duda@awf.krakow.pl (H.D.); tadek@ambrozy.pl (T.A.); 2College of Medical Sciences, Institute of Physical Culture Studies, University of Rzeszow, 35-310 Rzeszów, Poland; wojciechczarny@wp.pl; 3Department of Sport, University School of Physical Education, 51-612 Wrocław, Poland; Wieslaw.judo@wp.pl; 4Department of Physical Education and Sports, Matej Bel University in Banská, Bystrica, 974-01 Banská, Bystrica, Slovakia; gornerk@uek.krakow.pl

**Keywords:** city smog, monitoring, sports training, physical performance

## Abstract

The essence of a sports training includes not only developing the skills necessary in a chosen sport but also particular care about athlete’s health. This issue should be taken into account especially in case of children and youth engaged in sporting activities. In the paper there are issues connected to the control of physical effort abilities in the sports training of young football players and the assessment of the reaction of the body to physical exercise in city smog conditions (the environment of the city of Kraków) and clean air conditions (the environment of the town of Głuchołazy). This paper shows that, when assessing physical effort, one can consider not nly the results of physical tests but also the reaction of the body to a given physical load. One should remember that physical load depends not only on the methods used and the range of intensity, but also on the environmental conditions, like the quality of the air. Determining the reaction of the body to physical load (performance tests), taking into account the conditions in which the training takes place, prevents overloading and sets directions for rational sports training. The analysis of the results of the study leads to three main conclusions: (1) The planning of sports training has to consider not only the methods and means of the training but also environmental factors (air pollution); (2) Physical effort in smog conditions should be done with the use of antismog face masks; (3) The arrangement of sports training (particularly for youth) should strictly take into account the environment in which the training takes place.

## 1. Introduction

Common sports training is conducted in the conditions of using maximum of possibilities of an athlete’s organism, often without respecting pro-health rules. However we need to remember that the basic goals of undertaking physical activities include not only learning motor skills but above all the improvement of body functionality in the psychophysical domain, e.g., physical and mental health. Sports particularly for children and youths are basically activities for reviving (agitating) and reinforcing (improving functionality of the body). Therefore, the rational training of beginners should be pro-health, adequate for young developing bodies [1]. This should be taken into account when defining rational sports training—it results from the health doctrine itself (as in reinforcing the upbringing through sport). It should also be explicitly stated that when we practice sport activities we rarely pay attention to external conditions (e.g., climatic, including, above all, air quality) in which the exercise takes place [2].

In this paper, the air quality during the sports training of young football players is the main topic of consideration. Lately, this problem has been discussed in detail in Poland, where it has been attracting more and more interest. It turns out that in many countries [3,4], including Poland [5,6,7], air pollution in metropolitan areas considerably exceeds the norm set out to be safe for human health or life. Smog is a big problem for many Polish cities and towns. Thus, the question arises as to whether sports training in smog conditions is still pro-health.

Clean air contains almost 80% nitrogen, less than 20% oxygen and 1% carbon dioxide, as well as some other tracers. Although nitrogen is mostly just a solvent for oxygen, it is necessary for the production of proteins in plants [8]. Clean air has a great impact on our health—it influences the regular metabolism of our bodies. The importance of oxygen for our health can be confirmed by the fact that we can survive several days without food and water but only several minutes without air to breathe [9]. The human body is aerobic. We need air to live and we need clean, fresh air to keep our health [10].

Presently, clean air can be a challenge in major urban centers, which are often connected with industrial agglomerations [11], and this fact is shown in Figure 1.

Problems connected with greater and greater pollution create major threats to human health [12,13,14]. Inhaling polluted air, combined with an unhealthy lifestyle, are the main reasons for the increase in the occurrence of diseases of affluence, such as heart disease, diabetes, cancer, obesity and disorders of the immune system [15]. Polluted air also affects the endocrine system and mental health. It also causes an increase in the occurrence of asthma and allergies in children. According to the statistics of the World Health Organization, almost 20% of all deaths in Europe are caused by environmental diseases [16].

Taking this into account, the study presented in this paper is concerned with the assessment of the reaction of the human body during sports training to the conditions of the air. The study included young athletes who addressed training in an emotional manner (great motivation—great activity). The choice of young football players was intentional, as this sport is the most popular in Poland and the majority of youth sports clubs is unfortunately located in cities. The reactions of young athletes to physical exercise were determined based on physical performance, which principally depends on the efficiency of oxygen absorption by the human body [17].

Physical performance means the ability to endure hard or long-lasting physical exercise done with large groups of muscles without rapidly increasing fatigue. Changes in the body caused by motor activity (fatigue) are the reason for changes in the internal environment of the body. Performance also includes the tolerance of fatigue changes and the ability to quickly eliminate fatigue after exercise [18].

The duration of the exercises with steady or increasing intensity (including running, cycling (on a bike or a cycloergometer) and long-lasting walks in the conditions of constant homeostasis; “steady study”) is the real measure of physical performance.

The ability of the human body to absorb oxygen, the so-called oxygen uptake (VO_2_ max) or the aerobic capacity of the body, is the best known indicator of physical performance. This indicator allows for the prediction of the reactions of a healthy body to physical exercise in a wide range [19].

Physical performance can be characterized by the ability to do exercises with big general energy costs, not necessarily to do specific exercises or activities. A man achieving a high physical performance does not have to have more than an average capacity of the motor system.

Physical performance can be assessed by:The efficiency of the functions contributing to covering the oxygen demand of the muscles and the activity of biochemical processes in the muscles deciding on using oxygen sources of energy.The resources of energy substrates in the muscles and other tissues and the efficiency of mobilizing substrates from the sources outside muscles.The efficiency of the processes equalizing changes in the internal environment of the body caused by the exertion.The tolerance of fatigue changes.

Activities done in one’s spare time, except physical performance, should also improve the stamina, which is closely connected to performance.

Taking into account the above aspects of physical performance important to the health and stamina of the body, the change (improvement) in the parameters of performance due to regular physical activity taken in an environment with a different (better) quality of the air was the main object of this study.

For purposeful and rational sports training, it is necessary to define and learn the functional possibilities of the body during training, answering the given training load and to comprehensively determine the state of its current adaptive mechanisms responsible for exercising disposition [20]. Thus, this is about control in the field of physical performance, hence the assessment of the ability of the body to do a specific kind of physical work, given in terms of the level of the maximal efficiency of the effort, as well as the smooth running of the renewal processes [19]. The explanation of the essence of the mechanisms of exercising disposition in sports training is important since in the traditional control proceedings, this is often treated superficially or incorrectly—the conditions of exercise are not considered properly, e.g., smog is not taken into account. Moreover, the importance of the problem is stressed by the fact that global environmental pollution is becoming one of the most threatening problems of the modern world, like AIDS or Ebola virus [21]. Air pollution as a threat to athletes (absorbing pollution by the body during exercising) and should especially be taken into account in Poland [6]. Our country has the most smog-polluted air among all countries of the European Union [22]. 

Although the above figure presents air pollution in European countries according to norms from 2012, currently in Poland they stay on similar level, which causes a huge threat to health [22]. The fact is also confirmed by the position of the European Commission, which, in 2015, referred the matter to the European Court, claiming that the permissible daily PM10 content in the air was constantly exceeded in the majority of monitored regions in Poland (in 35 out of 46). The results of the reports of the WHO also claim that the reason for ca. 50,000 deaths in Poland is the smog conditions we live in [23,24]. It can be noted that increased absorption of the air by the body is typical for physical exercising and unfortunately is not always noted in the aspect of healthy sports training. Similarly, sports training is usually understood as exercising with an optimal physical load, but generally the environmental conditions are not considered.

The main goal of this study is to determine the direction of rational sports training, which should optimally be pro-health. Taking into account the fact that most training athletes do their exercises in an urban environment (numerous sports clubs) with high concentrations of air pollution [25], the paper’s goal was to determine the reaction of the human body to exercise with regard to the level of air pollution.

In the study process, the objective and subjective methods of the control of physical performance for the purpose of control in organized sports training are shown. To meet the study goals, the following questions were set:Does the impact on the parameters of physical performance depend on the training measures used or is it necessary to consider the environmental conditions connected to the level of air pollution in the planning of physical load?Does sports training in poor environmental conditions (high air pollution, smog) satisfy pro-health requirements?

Taking into account the main assumptions of pro-health training and sport activity in urban agglomerations, the following hypotheses were set:The increase in the indicators of physical performance depends not only on regular sports training but also on the environmental conditions—the quality of the air—in which the training takes place.Sport activity in polluted air (smog) not only can decrease the dynamics of the increase in performance but can also negatively influence human health.

## 2. Materials and Methods

The study included 30 young (aged 15–16) male football players, who were students of the School of Sports Champions—Football in Kraków (SMS-PN Kraków—Szkoła Mistrzostwa Sportowego—Piłka Nożna w Krakowie) in categories U-15 and U-16. The study was conducted periodically from 2015 to 2016. The study was approved by the Ethics Committee at the regional medical chamber in Krakow number: 42/KBL/OIL/2015.

Young athletes practiced regular sports training. Both the training and the study were conducted in a monthly mesocycle in two different environmental conditions due to the level of air pollution. The first one—urban— was set in the facilities of SMS-PN Kraków in the center of Kraków, where the air pollution is considerably high, [26,27,28,29] (latitude 50°03′41″ N, longitude 19°56′11″ E). Kraków has 12 points on the scale of pollution from 1–14, where 1 is the smallest amount of pollution and 14 is the largest and it is often considered to be the capital of Polish smog. The mean temperature in Krakow during the young footballers’ training was between 0° and 1° Celsius. 

These facts are also confirmed by a detailed study of the environment of the city of Kraków, in which the air pollution highly exceeds permissible norms.

The other environment is the small town of Głuchołazy, located between the Opawskie Mountains (part of Sudety), the Paczkowskie Foothills (in the Sudety Foothills) and the Głubczycki Plateau, by the River Biała Głuchołaska (latitude 50°18′54″ N, longitude 17°23′00″ E). The environmental conditions of the town include a pro-health microclimate, beneficial mostly for respiratory and circulatory systems [30,31]. Głuchołazy has only 1.5 points on the scale of pollution. The mean temperature in Głuchołazy during the young footballers’ training was between 0° and 1° Celsius.

In a training microcycle (weekly), there were 10 90-min sports training sessions. They consisted of warm-ups and the main part, in which continuous, interrupted and interval methods were used to train motor skills. According to the data from the monitoring of physical load, the training sessions were similar to one another in duration and intensity (the mean value of the intensity was mixed transition zone) [8]. The program of the above-mentioned Sports training was standardized (it contained the same training material). It was conducted for 12 days in Kraków (20 units)—1st half of January in 2015 and 2016 and for 12 days in Głuchołazy (sports camp---20 units)—2nd half of January in 2015 and 2016.

To find out how the exhaustion mechanisms developed due to the same physical load but in different environmental conditions (the quality of the air), a test for studying the performance and stamina of the body was applied. This was the Cooper test of physical fitness [32,33], which was conducted before and after the completed sports training cycle.

In order to assess the reaction of the body to physical effort to a greater extent, a 6-min walk test (6MWT) [34] was also used immediately after the Cooper test. Reaction of the organism was examined using Borg’s Rate of Perceived Exertion scale [35]. The sports training sessions and the Cooper test in both the urban environment of the city of Kraków and the small town environment of Głuchołazy took place on a track of a stadium. The program of the study is listed below:

Kraków 1st measurement: 3 days before completing the first sports training cycle in Kraków.

Kraków 2nd measurement: 3 days after completing the first sports training cycle in Kraków.

Głuchołazy 1st measurement: 3 days before completing the first sports training cycle in Głuchołazy.

Głuchołazy 2nd measurement: after completing both sports training cycles in Głuchołazy.

Kraków 3rd measurement: after 6 days in Kraków.

The differences in days between the consecutive measurements were as follows: 6 days between Kraków 1 and Kraków 2, 9 days between Kraków 2 and Głuchołazy 1, 9 days between Głuchołazy 1 and Głuchołazy 2 and 6 days between Głuchołazy 2 and Kraków 3.

To assess the quality of the air, an air quality meter—Xiaomi Mi P— as well as an application to check the quality of the air in Polish cities (“Kanarek”), which collects data from official measurements by the Chief Environmental Care Inspectorate (GIOŚ—pol. Główny Inspektorat Ochrony Środowiska), were used.

According to the data, the mean values of the air quality in the city of Kraków proved mediocre and showed poor air quality (PM_10_ = 124mg/m^3^, PM_2,5_ = 89 mg/m^3^, NO_2_ = 99 mg/m^3^, NO = 118 mg/m^3^, CO = 1290 mg/m^3^). However, in the town of Głuchołazy, the mean values of the air quality proved to be good and showed very good air quality (PM_10_ = 43 mg/m^3^, PM_2,5_ = 21 mg/m^3^, NO_2_ = 66 mg/m^3^, NO = 69 mg/m^3^, CO = 889 mg/m^3^). These ingredients, if exceeding the standards, significantly threaten human health [29,36].

In order to analyze the results of the study, basic statistical computations were used: mean value, standard deviation and a *t*-test, which was used to determine the level of the significance of the differences. The relation between the measured values was also studied with the use of Pearson’s correlation coefficient [37].

## 3. Results

Presenting the results of evaluating the stamina and performance parameters of young football players participating in an organized sports training in similar environmental conditions of air quality between the first and second measurements in Kraków (Table 1 and Table 2), one can notice a similar state of training efficiency—there are no statistically significant differences. Of course, the similar state of training efficiency can also be explained by the short time of training between the first and second measurements in Kraków, where, according to [9], functional changes in the body have not yet occurred.

Similar results were obtained in the analysis of the subjective assessment of fatigue measured after the Cooper test in the first two measurements in Kraków, using the Borg scale (Table 2). 

As can be seen, there were no significant differences between the results of the 1st and 2nd measurements, which can also prove the similarity of the environmental conditions (air quality and climate) in both measurements.

There are interesting changes in the results of the third measurement, done after 3 days’ stay in the training camp in Głuchołazy. Although in the results of the Cooper test the differences between the 2nd measurement in Kraków and the 1st in Głuchołazy are not significant (Table 3), the improvement in the results in the 1st measurement in Głuchołazy can be noticed.

Confirmation of changes taking place in the body of young athletes can also be found on the basis of research on the feeling of subjective fatigue according to the Borg scale [38]. We can see that the subjective fatigue noted during the Cooper test and Borg scale in Głuchołazy was less intense. In the comparative aspects of the obtained results we can, in a subjective opinion, notice significant differences in favor of staying in the environment of the city of Głuchołazy (Table 4).

The comparison of the next pair of measurements brings another proof of the main claim of the paper. The 2nd measurements in Głuchołazy were done after the young athletes spent 12 days in Głuchołazy. The analysis of the data presented in Table 5 shows a significant increase in the results of the Cooper test. The level of significance of the differences between the measurements is smaller than 0.01.

One can ask themselves what the main reason for such an increase in the results of the Cooper test was. The training load, according to the monitoring parameters (pulse rate), was similar, so it seems the choice of training methods and exercises was irrelevant. The quality of the air in Głuchołazy could be the differentiating factor between the results of both measurements. In order to confirm this statement, the results on the Borg scale in both measurements in Głuchołazy were compared. Data analysis (Table 6) shows a significant decrease in the subjective level of fatigue measured in the 2nd measurement in Głuchołazy.

The results of these measurements point out that physiological conditions for the physical effort of young athletes during their stay in Głuchołazy (due to good parameters of the quality of air) were more beneficial to their exercise than the ones in Kraków.

The next part of the study analyzes the results of the 2nd measurement in Głuchołazy and the 3rd measurement in Kraków. Again, in order to prove the impact of air quality to sports training, both tests’ (Cooper and Borg) results were compared.

The analysis shows that the results of the Cooper test measured in Kraków (3rd measurement) were lower than those measured in Głuchołazy (2nd measurement) (Table 7). While the differences between the measurements were not significant, the results of the 3rd measurement in Kraków decreased in comparison to the results of the 2nd measurement in Głuchołazy. One can think that the lower quality of the air in the Kraków environment (more toxic air due to smog) could have had an impact on the measured parameters of the young athletes [6].

The further analysis of the results on the Borg scale also confirmed a positive influence of the good air quality on the physical performance of the body (Table 8).

The analysis of the study results clearly shows that the increased fatigue after exercising when comparing the 2nd measurement in Głuchołazy and the 3rd measurement in Kraków could have resulted from worse environmental conditions and the lower quality of the air, which caused worse wellbeing in the young athletes.

## 4. Discussion

The sports training of children and youth should be rational in the aspect of choosing proper physical load. It is important due to pro-health impact of the training which requires not only choosing proper methods but also appropriate quality of the air in the training environment. In the study assumptions, sports activity and its importance for the improvement of the general physical fitness of the young football players were reviewed. It is certain that, together with increasingly polluted air (smog conditions), a great amount of toxic substances enter the human body. This, of course, broadly poisons the body’s tissues [8].

Thus, this study concerns the importance of optimal physical effort (enhanced sports activity) and its impact on the physical performance of young bodies. Optimal physical effort means not only the best conditions of the duration and intensity of training but also takes into account the environmental conditions, like the quality of the air during the training [20]. Sports training is more popular in big cities due to the number of sport schools and clubs. However, it often takes place in a polluted environment (city or industrial smog), which poses a threat to human health. This fact was confirmed by a Chinese study, which proved that air pollution has a negative impact on health when people are physically active [39]. These forms of sports activities do not meet pro-health standards, they can even have harmful effects to health [40]. One of the goals of sports training is the increase in the physical performance of an athlete. This study showed that this parameter was dependent not only on the optimal duration and intensity of the training but also on the environmental conditions, including the quality of the air. People who have better physical performance can function better and more economically. They are more productive and have better health. When seeking methods that increase the physical performance of the body, we should undertake rational activities which yield tangible benefits for our sport results, but most importantly, to our health. This problem is extremely important, especially in the training of children and youths, whose bodies are developing and it has a huge impact on their both physical and mental health.

This study showed significant physiological changes under disadvantageous conditions of the environment. Thus, rational sports training should consider not only optimal individual training load but also adequate climate conditions—clean air [6,41]. This aspect is often unnoticed by athletes, hence, one can often see youths training in big cities where the quality of the air is very poor. A good example of this was the Olympic Games in Beijing in 2008, which took place when the air in the city was extremely polluted [42]. Such conditions usually have a negative impact on health, especially the health of young people, and together with increased physical effort, the body absorbs harmful substances from the air [5,43]. A study showed that a high concentration of pollutants during physical activities slows cardiovascular functions, as well as hematological parameters [44]. There should be just one solution in this situation—physical effort in conditions of polluted air should be radically restricted. Such a phenomenon was observed in this study—worse results (ineffective physiological reactions) due to unfavorable conditions and worse wellbeing was observed during the training in more polluted air.

Thus, taking into account existing problems with changing this attitude (educational issues) for undertaking healthier sport activities, it seems that the only alternative for sport activity of mostly young people in smog conditions is using antismog face masks, which efficiently filter polluted air.

The problem of rational physical effort in city or industrial smog conditions is very important to our health. The analysis of the data shows that undertaking sport activity in smog can be dangerous because during enhanced physical activity, metabolism and breathing rate accelerate and ventilation increases. As a result, we breathe almost 20 times more air than during a walk, and with it harmful chemical substances [8], which can do serious damage to our health. In Poland, this topic has been discussed more and more but unfortunately no specific guidance for participants in sport activities has been created. It seems that physical activity participants should know the level of pollution of their region and instructors and coaches should take this into account when setting the duration and intensity of the effort. The training itself should be done with the use of antismog face masks.

In summing up, it should also be said that rational sports training should act on the body of the athlete in a mild manner, so an assessment that is based on only the stamina of the athletes (often used in traditional training) can be misleading. Taking into consideration the reaction of the body to physical load (performance tests, wellbeing) in the given environmental conditions (the quality of the air in the region) prevents overloading and can set directions for individual and pro-health training in sport activity.

## 5. Conclusions

Choosing the physical load in sports training has to consider the reactions of the body to physical effort.The planning of sports training has to consider not only the methods and means of the training but also environmental factors (air pollution).Due to health issues, sports training in a city in smog conditions should be significantly reduced.Physical effort in possible smog conditions should be done with the use of antismog face masks. Masks should be mandatory when the air pollution is particularly adverse.The arrangement of sports training (particularly for youths) should strictly take into account the environment in which the training takes place.

## Figures and Tables

**Figure 1 ijerph-17-05510-f001:**
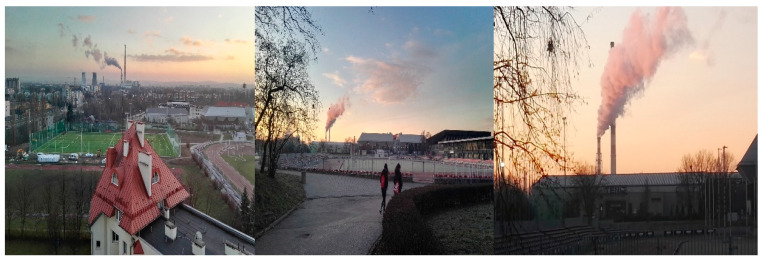
City smog in Kraków (source: authors’ archives).

**Table 1 ijerph-17-05510-t001:** The values of the stamina and performance parameters of the football players in the 1st and 2nd measurements in Kraków.

Measurement Parameters	Cooper Test Kraków 1st Measurement	Cooper Test Kraków 2nd Measurement
Mean	2716,67	2731,00
Standard deviation	130,89	131.35
Coefficient of variation	4.82	4.81
Significance of the differences	0.3364	

**Table 2 ijerph-17-05510-t002:** Assessment of the subjective level of fatigue of the tested individuals in the Borg scale for the first and second study in Krakow

Measurement Parameters	Borg scale Kraków 1st Measurement	Borg scale Kraków 2nd Measurement
Mean	16.93	16.80
Standard deviation	0.83	1,03
Coefficient of variation	4.89	6,13
Significance of the differences	0.291	

**Table 3 ijerph-17-05510-t003:** The values of the stamina and performance parameters of the football players in the 2nd measurement in Kraków and the 1st measurement in Głuchołazy.

Measurement Parameters	Cooper Test Kraków 2nd Measurement	Cooper Test Głuchołazy 1st Measurement
Mean	2731,0	2752,67
Standard deviation	131.35	131.46
Coefficient of variation	4.81	4.78
Significance of the differences	0.262	

**Table 4 ijerph-17-05510-t004:** Assessment of the subjective level of fatigue of the tested individuals in the 6MWT for the second study in Krakow and the first study in Głuchołazy.

Measurement Parameters	Borg scale Kraków 2nd Measurement	Borg scale Głuchołazy 1st Measurement
Mean	16.80	16,23
Standard deviation	1,03	1,14
Coefficient of variation	6,13	6,99
Significance of the differences	0.0238	

**Table 5 ijerph-17-05510-t005:** The values of the stamina and performance parameters of the football players in the 1st and 2nd measurements in Głuchołazy.

Measurement Parameters	Cooper Test Głuchołazy 1st Measurement	Cooper Test Głuchołazy 2nd Measurement
Mean	2752	2833.0
Standard deviation	131.46	115.68
Coefficient of variation	4.78	4.08
Significance of the differences	0.0074	

**Table 6 ijerph-17-05510-t006:** Assessment of the subjective level of fatigue of the tested individuals in the 6MWT for the first and second study in Głuchołazy

Measurement Parameters	Borg scale Głuchołazy 1st Measurement	Borg scale Głuchołazy 2nd Measurement
Mean	16,23	15,40
Standard deviation	1,14	1.07
Coefficient of variation	6,99	6,95
Significance of the differences	0.0025	

**Table 7 ijerph-17-05510-t007:** The values of the stamina and performance parameters of the football players in the 2nd measurement in Głuchołazy and the 3rd measurement in Kraków.

Measurement Parameters	Cooper Test Głuchołazy 2nd Measurement	Cooper Test Kraków 3rd Measurement
Mean	2833,0	2815.17
Standard deviation	115.68	122.85
Coefficient of variation	4.08	4.36
Significance of the differences	0.282	

**Table 8 ijerph-17-05510-t008:** Assessment of the subjective level of fatigue of the tested individuals in the Borg scale for the 2nd study in Głuchołazy and the 3rd study in Krakow.

Measurement Parameters	Borg scale Głuchołazy 2nd Measurement	Borg scale Kraków 3rd Measurement
Mean	15,40	16,07
Standard deviation	1.07	0,98
Coefficient of variation	6,93	6,10
Significance of the differences	0.007	

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
