# Peer review of "Reaction of the Organisms of Young Football Players to City Smog in the Sports Training"

_ijerph, 2020, doi:10.3390/ijerph17155510_

Round 1

Reviewer 1 Report

As a reviewer I have the following remarks

  1. Line 48, I suggest (Source [7]). – no bold, new line.
  2. Line 61: City smog [12-14]. (Fig. 2).
  3. Line 69. “the reaction of human organism during sports training to the conditions of air”; You only consider air pollution. It will be good to mention and include temperature in Krakow and Glucholazy. Say, provide average for the period of the study in two locations.
  4. Line 116: You have many “[*]” in italic, for example “taken into account in Poland [6].”
  5. Line 119: The label on the fig. It’s “1ng/m^3”. It looks as “lng”, should be “1 ng/m^3” – a space after 1.
  6. Line 151: “The study included 30 young” – please provide age range, boys?
  7. Line 157:” air pollution is considerably high [26.27,28] Kraków..” why not [26-28]?
  8. Figure 5, we need the used units for PMs and NO2, also I think in the text you need provide (mg/m^3 etc). Say, you may add in the Fig caption.
  9. Line 165: “a small town Głuchołazy “- consider to provide geographical coordinates for the both locations. Many authors do this, as some local knowledge is not an universal.
  10. Line 181: “always three days before and three days after completed sports training cycle” and after in the points you specify various time schedule. In this context I don’t understand “6 days before 2nd measurement”. Think about re-wording. The distance between measurements is minimum 7 days (including the training days as 0)?
  11. Line 196: “CO” and also units.
  12. Line 201: “mean value, standard deviation and t-test” – Have you used the paired t-test? Explain. You compare the means from a few various treatments (locations).
  13. The values in your table, please considered to use only one digit of accuracy. Say, replace “2714.17” by rounded 2714.2.
  14. Your tables –then can be simplified and compact, you can merge them to reduce their numbers. Using some abbreviations, it is possible to put more columns. Now the main body of the tables is a long text as “Kraków 2nd measurement “, could be just K2. Think about it. It may stay, but also think about potential readers.
  15. There are some publications on air pollution and fatigue, also the Air Quality Health Index is used to prevent excess exercises outdoor when air quality is bad.
  16. Please read the paper again.

Thank you.

Author Response

Thank you for reviewing our paper and for all suggestions. We took them into account and made suggested changes.

  1. Corrected
  2. Corrected
  3. We added the mean temperature in both locations.
  4. Corrected
  5. Corrected
  6. We added age and sex of the players (age was also given in categories U-15 and U-16)
  7. Corrected
  8. Units added
  9. Geographical coordinates of both locations were added.
  10. The schedule of training routines was written in more understandable way.
  11. Corrected
  12. The t-test was used. Statistical significance of the differences between the means is included in each table.
  13. We rounded the values in the tables.
  14. It is a good idea, yet we would like to stay with our tables – we think it is clearer and more understandable.
  15. We checked the details of the article.

Reviewer 2 Report

Manuscript ID: IJERPH –

Title: "Assessment of the reaction of the organism to city smog (on the example of the analysis of parameters of physical performance in the training of young football players"

The aim of this manuscript is to analyse the effect of city smog on sport performance in young football players. To this aim, 30 young football players were enrolled in the research and tested in different places: Krakow (high smog) and Glucholazy (low smog). Overall, data seem to indicate a very bad effect of smog on performance.

The topic of the manuscript is potentially interesting. However, I have several concerns on the present form.

Title is too long.

Introduction is a little big long but clear and well documented. Illustrations are very nice even if the connection with the text is not always clear (see for example figure 4).

The experimental schedule is not clear. Why are test sessions performed after 6 days in Krakow, and just after 3 days in Glucholazy?

In the results section, Authors wrote about statistically significance or not significance, however no statistical test is quoted. Which kind of analyses did Authors perform?

Discussion section could be shortened, and the take home message made more explicit.

Author Response

Thank you for reviewing our paper and for all suggestions. We took them into account and made suggested changes.

  1. We shortened the title to Reaction of the organisms of young football players to city smog in the sports training.
  2. We added more details describing fig.4 in the introduction.
  3. The description of the schedule was modified. The timing of measurements in Głuchołazy depended on the duration of the camp (12 days). The first one was done after 3 days in order to acclimatize, the second one was done the last day of staying which was 9 days after the first measurement. In Kraków there were always 6 day between the measurements (a weekly microcycle). The time between the measurements was 9 days (Kraków 2 – Głuchołazy I and Głuchołazy I – Głuchołazy II).
  4. The t-test was performed and the statistical significance of the difference between the means is included in each table.
  5. The discussion was shortened and it should be clearer.

Round 2

Reviewer 2 Report

The revised version of the manuscript is improved